# Predicting Relapse in Substance Use: Prospective Modeling Based on Intensive Longitudinal Data on Mental Health, Cognition, and Craving

**DOI:** 10.3390/brainsci12070957

**Published:** 2022-07-21

**Authors:** Anders Dahlen Forsmo Lauvsnes, Rolf W. Gråwe, Mette Langaas

**Affiliations:** 1Department of Mental Health, Faculty of Medicine and Health Sciences, Norwegian University of Science and Technology, 7491 Trondheim, Norway; rolf.w.grawe@ntnu.no; 2Kvamsgrind Addiction Treatment Centre, 7036 Trondheim, Norway; 3Clinic of Substance Use and Addiction Medicine, St. Olavs University Hospital, 7006 Trondheim, Norway; 4Division of Psychiatry, Department of Research and Development, St. Olavs University Hospital, 7006 Trondheim, Norway; 5Department of Mathematical Sciences, Faculty of Information Technology and Electrical Engineering, Norwegian University of Science and Technology, 7491 Trondheim, Norway; mette.langaas@ntnu.no

**Keywords:** substance use disorder, prospective modeling, cognition, craving, mental health, relapse

## Abstract

Patients with severe substance use disorders are often characterized by neurocognitive impairments and elevated mental health symptom load, typically associated with craving intensity and substance use relapse. There is a need to improve the predictive capabilities of when relapse occurs in order to improve substance use treatment. The current paper contains data from 19 patients (seven females) in a long-term inpatient substance use treatment setting over the course of several weeks, with up to three weekly data collections. We collected data from 252 sessions, ranging from 1 to 24 sessions per subject. The subjects reported craving, self-control, and mental health on each occasion. Before starting the repeated data collection, a baseline neuropsychological screening was performed. In this repeated-measures prospective study, the mixed-effects models with time-lagged predictors support a model of substance use craving and relapse being predicted by the baseline reaction time as well as the temporal changes and variability in mental health symptom load, self-control, and craving intensity with moderate to high effect sizes. This knowledge may contribute to more personalized risk assessments and treatments for this group of patients.

## 1. Introduction

Substance use disorders (SUD) are relapsing in their nature, with relapse rates of about 40–60% in general [1] and about 30% in the first year after inpatient treatment [2] To better understand the reasons and precursors of relapse in substance use disorders, much research is being undertaken into understanding the underpinnings of maladaptive decision-making patterns in this clinical population [3] In contrast to the knowledge about the trait risk factors of relapse, the dynamics of the mental and behavioral factors preceding and influencing the likelihood of imminent relapse prospectively are less understood [2]. The probability of relapse is predicted by craving [4], executive self-control [5] and mental health distress [6]. The associations between craving and actual substance use are complex and vary [4]. Self-control, defined as the ability to prioritize long-term goals over short-term gratification or relief, has, in some instances, been found to moderate the relationship between craving and alcohol use [7], but not consistently for illicit drug use [8]. Nevertheless, diminished behavioral self-control may, combined with heightened incentive salience for substance use, perturbate substance use behavior and as such, subjects with impaired self-control are often prone to relapse [1] and poor treatment retention [9].

Executive functioning, in general, and self-control, is influenced by different internal and environmental factors such as sleep, mood, and stress, and hence the ability to perform self-control varies across situations and individuals. Self-control has typically been considered a stable construct across time (trait). Still, there is increasing awareness of its temporal variability [10], and how fluctuations in self-control functions are related to the degree of craving, mental distress, and substance use behaviors in general [11,12]. There is also a well-established association between internalizing mental health symptoms and substance use. This relationship is not fully understood, but coping (reducing negative emotions) and enhancement (increasing positive emotions) may link the two [13]. Executive functioning also influences the effects of internalizing symptoms on substance use behavior [10,14] by affecting the ability to regulate mood [15]. 

There is an increasing interest in improving our understanding of the effects of within-subject variability in predictor variables such as self-control in substance use research [15]. However, many studies are limited by retrospective recall bias [15], and there is a void in the literature for prospective studies. Repeated momentary assessments enable a better understanding of the associations between craving, substance use, and their neurocognitive covariates [8]. These may help create a so-called biosignature of substance use disorder symptom deterioration and relapse [16]. Furthermore, it is reasonable to suppose that a better understanding of the associations between variability in measures of mental health, cognition, craving, and substance use in a naturalistic treatment setting may inform prevention work within substance use and enhance the personalization of assessment and interventions. 

The current study aims to explore the prospective relationships between mental health variables, executive self-control, substance use craving, and relapse in a sample of polysubstance use inpatients. We believe that this will improve the quality of predictions compared to current research by improving the understanding of the temporal co-variability of these phenomena and thus improve our ability to provide personalized, just-in-time interventions to this group of patients. 

## 2. Materials and Methods

### 2.1. Participants

The participants in this study were enrolled from an inpatient population at a long-term therapeutic community addiction treatment center in Norway (3–12 months of hospitalization) and were between 18 and 27 years old. The participants were included as part of a prospective cohort study exploring the feasibility of working memory training in a polysubstance use population with dual diagnosis. Participants were included no earlier than one month from their initial admission to inpatient treatment. After obtaining informed consent, the participants underwent baseline assessments for mental health and neuropsychological performance before the data collection was started. Following the inclusion assessment of baseline data, 19 patients were enrolled in the study, and data from 252 individual sessions were collected in addition to the baseline data. Data were collected in parallel to the working memory training sessions. The participants enrolled attended a mean of 13.3 sessions (median 12, range 1–24), with a minimum of 2 days between sessions. One participant reported zero sessions after the inclusion assessment and consequently was excluded from the repeated measures analyses but remained included in the baseline data presented here. 

### 2.2. Measures

All baseline measures were obtained at the same occasion. Participants underwent interviews and neuropsychological testing in a neuropsychological lab setting, with sufficient space and sound insulation to avoid disturbances. Repeated measures were obtained either in the lab, or later, due to COVID-19 distance restrictions, in a meeting room.

### 2.3. Baseline Measures

***WAIS-IV.*** The Wechsler Adult Intelligence Scale-Fourth Edition (WAIS-IV) measures the cognitive abilities of individuals between 16 and 90 years and 11 months of age. We used a 7-subtest version proposed by Meyers, Zellinger, Kockler, Wagner, and Miller [17]. However, we did not estimate the Full-Scale IQ from the abbreviated administrations due to our limited sample size and the lack of validation of such estimation in the Norwegian version of the WAIS-IV. We report on two composite scales, the Working Memory Index and the Verbal Comprehension Index, and scaled scores of the additional subtests from the baseline assessment of the participants. The measurements are referred to as WAIS and the respective subtests and indices. 

***D-KEFS.*** The Delis-Kaplan Executive Function System (D-KEFS) is a neuropsychological measure for use with adults and children aged 8–89 years old. It is used to measure a variety of verbal and nonverbal executive functions. It consists of nine separate tests that stand alone and thus do not provide aggregate measures or composite scores for performance. We used the Trail Making Test (TMT), Color-Word Interference test (CWI), and the Verbal Fluency test (FAS). The TMT measures the cognitive flexibility and speed; the CWI measures aspects of response inhibition. 

***CPT-3.*** The Conners’ Continuous Performance Test-Third Edition is a computer-based GO/NO-GO assessment of attention including impulsivity, inattention, sustained attention, and vigilance. 

Measurements were taken at the baseline and are referred to as WASI, D-KEFS, and CPT followed by the name of the respective subtests.

***Baseline diagnosis*.** All patients had severe substance abuse problems as a prerequisite for entering long-term inpatient SUD treatment, often combined with considerable psychiatric comorbidity. Baseline substance use disorder diagnoses (ICD-10) were obtained from the participant’s electronic health records and may be found in Table A1 in the Appendix A.

### 2.4. Repeated Measures

***SCL-5.*** The Hopkins Symptom Check List 5-item version (SCL-5) was performed at each session and used as a repeated input variable. SCL-5 has been used extensively in population studies in Norway and has a high psychometric validity [18] and reliability [19]. The scale has five items scored on a Likert scale from 1 (Not at all) to 4 (Very much). We calculated a mean score by dividing the total score by the number of items answered, giving a possible range of 1.00 to 4.00. The measurements are referred to as SCL.

***Self-control.*** Participants were asked to rate their subjective self-control on a visual analog scale measuring 10 cm, which was then measured and coded to the closest millimeter, giving a value on the 0–100 numerical interval. This procedure was conducted at each session. Single-item assessment of self-control has been found to capture cognitive self-control satisfactorily [20].

***Craving and substance use.*** Craving was assessed on a visual analog scale measuring 10 cm, and it was then measured and coded to the closest millimeter, giving a value on the 0–100 numerical interval conducted at each session. More elaborate psychometrical assessments of craving typically contain the theoretical components of desire, want, urge, and need [21]. Measures of craving need to be adapted to the context, taking into account the workload, sensitivity, and specificity [22], and as such, single-item visual analog scales are considered to have acceptable validity in measuring craving intensity [4]. We also recorded the actual substance use from the participant’s electronic medical health records based on documented self-reported drug use or positive urine samples post hoc and matched these to the sessions. These measurements are referred to as relapse. 

### 2.5. Statistical Analyses

We aimed to analyze a longitudinal dataset consisting of frequent, repeated measures data. The R statistical programming software was used for all statistical analyses [23]. Data were analyzed both at the aggregated group level (baseline and across repeated measures) and the within-subject level.

#### 2.5.1. Group Level Analyses

We created variables for the means and standard deviations (across all sessions, for each participant) and reported median and interquartile range (IQR) of the individual means and standard deviations across participants to describe the input and response variables in our dataset (Figure 1). Furthermore, the Wilcoxon–Mann–Whitney test was used to compare the means from two independent samples using the wilcox.test() function in R. The Pearson product–moment correlations were used to investigate the associations on the group level, and calculated using the cor() function in R. Testing whether a correlation was different from 0 was conducted based on the t-distribution with the corr.test() function in the Psych-package [24]

#### 2.5.2. Within-Subject Analyses

We used the scale ()-function in R to standardize the data (for each individual) to compare the repeated measurements with different scales. This function uses the mean and standard deviation over the entire data range (all sessions). We also visualized the individual variability in repeated measures data in the individual participant diagrams below using the ggplot()-function in the tidyverse package in R [25] (Figure 2). For all further statistical analyses, the unscaled raw data were used. We also created a rolling standard deviation variable for the individual for each SCL, self-control, and craving intensity measurement based on the three previous sessions to be used as input variables in the analyses (described below) at the within-subject level.

We calculated the intraclass correlation (ICC) to evaluate the effect of individual nesting on the outcome (self-control, craving, and relapse), where the ICC ranged from 0 to 1, and a higher ICC indicates a low data independence, hence the need for linear mixed models (LMM), since a substantial portion of the variance is then accounted for by within-subject factors [26]. An assumption of independence is invalid in repeated measures datasets since each participant provides more than one data point. One approach to overcome this is to aggregate the average repeated measures data across sessions and possibly also across participants, but this may not be optimal in the case of substantial individual differences [27].

To evaluate the within-subject effects of repeated measures on the outcome, we used linear mixed-effects methods and generalized linear mixed-effects models (GLMM) for the longitudinal (repeated measures) data. Measurement occasions were unevenly distributed regarding intervals and date, so occasions were ordered as sessions 1–24. In the LMM and GLMM analyses, the response was at time *t + 1* while the input variables were at time *t* (lagged input data) to allow for the prediction of the ‘next session’ outcome (*t + 1*) and only using data up to the previous session (*t*). Repeated measures were studied at level 1 (each participant), and all observations were nested within participants at level 2 (the observations in time for each participant) by including a random intercept where each participant received their own intercept [26]. We used the lmer() and glmer() functions from the lme4 package in R [28]. *p*-values from the models were extracted using the parameter package in R [29]. We used the craving, self-control, and relapse variables as response variables with a lead of one session. 

We used the full dataset (n= 19, baseline plus repeated measures, 252 sessions) for the descriptive statistics and between group comparison. In the analysis with mixed effect models, we removed all the data after each participant’s first substance use episode (leaving n = 17, and a total of 180 sessions) since a variability in symptoms may occur as a result of substance use [5]. In addition, we selected the baseline neurocognitive assessments to be included in the modeling. One advantage of mixed-effects (multilevel) models is that they do not require complex imputations but use all of the available data. As our models contained both random and fixed effects, we considered the conditional R-squared as a measure of variance explained, as implemented in the R package performance [30]. We also considered the AIC of each model, where a lower relative AIC indicates a better fit with less complexity [31].

## 3. Results

### Group-Level Analyses

Summary statistics of the baseline and repeated measures are available in Table 1 and Table 2. The sample consisted of 19 participants (seven females), with a mean age of 23.2 (SD = 2.2). Seven participants experienced one or more substance use episodes during the study period, for a total of 11 substance use episodes. 

Descriptive statistics of the baseline data with comparisons for the participants with or without substance use episodes in the study period indicate that the participants that experienced relapse had a significantly higher reaction time on CPT than the non-relapsing participants (*p* = 0.032) and that the baseline consistency in response time (CPT-HRT-SD) was higher in the relapsing participants (*p* = 0.0182). The baseline score of variability in response speed (CPT-HRT-SD) was negatively correlated to the mean of craving (−0.48, *p* = 0.050), indicating that less variability (i.e., higher t-score on the CPT HRT SD) was associated with lower craving intensity. However, the CPT reaction time itself (CPT-HRT) was not significantly correlated to the mean of craving (*p* = 0.309). We found limited associations between the baseline neuropsychological measures and repeated self-control measures.

To compare the participants with and without relapse in the study period on the repeated measures (across participants and occasions), summary measures (mean and standard deviation) were calculated across all sessions and for all participants (see Table 2), and we report here the median and inter quartile range for the craving, self-control, and SCL assessments of mental symptoms. With the exception of the mean of self-control, we found moderate effect sizes for relapse/non-relapse for all scores, although none were statistically significant (see Table 2 for the effect sizes for repeated measures). Nevertheless, we did find a tendency toward a significant group-level difference in the mean craving between relapse/non-relapse participants (*p* = 0.085).

The Pearson product–moment correlations for the aggregated scores across repeated measures (means and standard deviations over occasions for each participant) and the baseline CPT variability (CPT-HRT-SD) can be seen in Figure 1. Strong positive Pearson product–moment correlations existed between the standard deviations of repeated craving (craving SD) and perceived self-control (self-control _SD). (r = 0.89, *p* = 2.2 *×* 10^−6^). The variability in craving (craving SD) and mental health (SCLSD) was also moderate and significantly correlated (r = 0.50, *p* = 0.040). 

Figure 2 provides a graphical representation of the scaled values of the repeated measurements by session number per participant.

From a visual inspection of Figure 2, it is evident that craving, SCL, and self-control seem to vary both individually and considerably between participants, probably with high variability in within-subject means, suggesting effects on both between and within-subject levels. 

Intraclass correlations for all three outcomes, with a lead of one session, were calculated with separate mixed models with only an intercept term as a fixed effect (self-control *t + 1*, craving *t + 1* and relapse *t + 1*), resulting in ICC scores of 0.71, 0.48, and 0.21, respectively. Finally, we fitted the mixed-effects models using as the covariates mental health level (SCL) and variability (3-session rolling SCL-SD) and level (self-control) and self-control variability (3-session rolling self-control SD) and level (craving) and variability in craving (craving 3-session rolling SD). As responses, we used either self-control (self-control *t + 1*), craving (craving *t + 1*), or relapse (relapse *t + 1*) with a lead of one session. We also included the baseline variability in response speed on the CPT (CPT HRT SD) for the models trying to predict self-control and craving due to the significant between-group difference for relapse/no relapse. Both the CPT hit reaction time and variability (SD) of reaction time were significantly different between these two groups (*p* = 0.032 and 0.018, respectively). The resulting fits are shown in Table 3. All models achieved lower AIC scores than the respective intercept-only models for each outcome, indicating a better model fit.

We found that the level of self-control at time *t + 1* was predicted by the previous level (*t*) of SCL (*p* = 7.94 × 10^−4^), but not by the variability in SCL (*p* = 0.351) or the baseline measure CPT HRT SD (*p* = 0.493). Furthermore, the level of craving at time *t + 1* was significantly predicted by a 3-session rolling variability in self-control (*p* = 0.023) and CPT HRT SD (*p* = 0.033). Relapse was first analyzed in a model with 3-session rolling variability in craving, self-control, and SCL. Relapse at time *t + 1* was marginally predicted by rolling variability in craving (*p* = 0.061). In separate analyses of craving, self-control, and mental health (SCL) as individual predictors of relapse, we found that the 3-session rolling SD of craving (*p* = 0.020) and the level of SCL at *t* were significant predictors of later relapse at time *t + 1*. In addition, we found that levels of self-control, craving, and SCL were significantly predicted by their previous levels. We failed to include CPT HRT SD in the modelling of relapse *t + 1* (Model 3) since this provided a singular fit. The latter may be because craving and self-control are both included as predictors in that model and are correlated with each other (level and SD) and CPT HRT SD. To check for this, we ran models with only self-control SD, craving SD, and CPT HRT SD as predictors of relapse *t + 1*. This resulted in the craving SD still being the only marginal significant predictor (*p* = 0.073). 

To illustrate the effects of the significant predictor variables on the lagged outcomes, we calculated the predicted changes in the continuous variables (craving, self-control, and SCL) and the odds ratio for the binary output of relapse. Refer to Table 3 for a definition of models. Model 1’s outcome is self-control at *t + 1*. The mental health symptom (SCL) level across participants and time was 10.5 (see Figure A3 in the Appendix A for variability), for simplicity, we assumed that half of this level may be an interesting change and hence used a SCL change of 5.25. In the model for self-control *t + 1* in Table 3, if the SCL was changed by 5.25 units for one chosen participant, and the level of the rolling SD for SCL and the CTP HRT SD was kept constant, so the level of self-control has a predicted decrease by 2.10 × 5.25 = 11.0 units. For Model 2, the rolling standard deviation for self-control and the CPT HRT SD (baseline) were significant predictors for craving *t + 1.* For the rolling standard deviation for self-control, the level across participants and time was 9.8 (see Figure A4 in the Appendix A for variability,) and then half of this level was 4.9. Suppose we change the self-control rolling standard deviation with 4.9 units for one chosen participant, while keeping all the other predictors constant. In this case, we predict that the level for craving will increase by 0.55 × 4.9 = 2.7 units. For the CPT RT SD (baseline), the median level across participants was 50.9 and half was 25.45. If we change the CPT RT SD (baseline) with 25.45 units for one chosen participant, while keeping all of the other predictors constant, we predict that the level for craving will decrease by 1.08 × 25.45 = 27.5 units. Finally, for Model 4, the rolling standard deviation for craving is a significant predictor for relapse at *t + 1*. The rolling standard deviation for craving had a level of 14.0 across participants and occasions (see Figure A2 in the Appendix A for variability), and then half of this level was 7.0 and chosen as an interesting change. If we choose one participant and change the rolling standard deviation for craving with 7 units, the predicted change for relapse *t + 1* is an odds ratio of exp (0.10 × 7) = 2.0.

## 4. Discussion

We found that the mental health symptom intensity predicted self-control, that self-control variability predicts future craving intensity, and that variability in craving predicts later relapse. We also found that the baseline variability in CPT response time was a significant predictor of craving, but not of self-control. Although both repeated measures of self-control and mental health variables were predictive of craving intensity, our data showed that a model with only subjective self-control provided the best fit with the least complexity for predicting craving intensity and that the 3-session rolling SD of craving was the best predictor of relapse t + 1. As noted above, a realistic increase in the level of mental health distress decreased aa subsequent level of self-control with over 11%. Furthermore, increased variability in self-control increased the level of subsequent craving intensity, albeit modestly. Finally, a moderate change in craving variability had an important effect with an odds ratio change of 2.0 on subsequent relapse or a 100% increase in the risk, given this exposure. It also seems from our results that the effects of baseline variability in inhibitory control had a larger effect on craving intensity. These results indicate that mental health symptom intensity and both variability in self-control and baseline inhibitory control are important predictors of craving and subsequent relapse, with self-reported self-control being less influential than the baseline measure. 

The role of the affective state on impulsivity has previously been subject to thorough investigations as well as in the substance use population (i.e. [32]. Evidence suggests that increased mental health symptoms are related to decreased impulse control and hence has a detrimental effect on impulsive behaviors such as substance use [33]. Our findings support this, as we identified significant prospective relationships between mental health distress, self-control, craving, and relapse. Our findings also indicate a momentary relationship, meaning that mental health symptom load at time *t* predicts decreased self-control at the next measurement *t + 1*. In line with recent research [34] on the predictive relationship between mental health and self-control, the level of mental health symptoms and not variability predicted the subsequent self-control levels. This adds to the current knowledge from cross-sectional studies of substance use populations, where perceived self-control has been found to be associated with mental distress [32,35]. In addition, self-control outperformed mental health variables in predicting craving intensity and relapse status in time-lagged models in this sample, suggesting that self-control plays an important role as both targets for research, monitoring, and intervention. However, this effect was modest, and it seems that the baseline inhibitory control had a larger effect on subsequent craving intensity levels. This has previously primarily been found in longitudinal studies with fewer measuring points than our study (i.e., [11]). Therefore, our approach with higher frequency data supports previous findings and fills a void in the existing literature by providing increased data granularity. McKee et al. [34] pointed out that using measures of variability in repeated experience sampling sessions, adds predictive performance in substance use research, and our findings support their findings. Furthermore, the current study occurred in a naturalistic inpatient treatment setting for young adults with substance use disorders and often dual diagnoses. We measured mental health frequently using a valid psychometric tool without manipulating the patient’s affective state in any way. This lack of experimental condition means that fluctuations in the affective state were due to intrinsic or extrinsic factors outside the influence of the researchers. Other studies have identified the effects of mental health on craving. It may, however, be the case that the lack of experimental manipulation of mood may explain the difference in the effect of mood on craving between our and other studies.

This is one of the first studies assessing the relationships between mental health, cognition, craving, and relapse in a naturalistic clinical setting in an intensive, prospective design. In combination with many observations per participant over time, we believe that these are strengths of the current study that may justify the moderate number of participants. 

Due to the small sample size, and the common clinical characteristics of patients in long-term inpatient substance use treatment, we did not perform drug-specific analyses. Other studies that have conducted analyses between drugs have identified some differences [8] so this might have been a drawback of our approach. It cannot be ruled out that moderate/smaller, but nevertheless clinically important effects, were not discovered in the current study and that effects may be more reliably estimated with a larger sample. We chose the 3-session rolling standard deviation length based on available data and some previous studies. However, this may seem a bit arbitrary. With a larger dataset, it would have been interesting to compare different intervals of these variability measures and hence the prediction window. Another valid criticism of the current study may be that the participants took part in a working memory training program and that this could have been related to their experience of self-control. We do not know if such an effect is similar for all participants, but the effect of such training has previously been shown to be modest in substance use populations [36]. Although we found a significant effect of self-perceived self-control on craving intensity, this effect was modest, especially compared to the baseline performance measures of inhibitory control. This may also be due to the measure itself; hence, tools that better capture inhibitory control might provide better insight. Further research should investigate the measures of impulsivity and mood in outpatients with severe substance use disorders. Research designs that allow for higher data granularity and possibly even digital proxies of variability in neurocognitive, mental health, and substance use-related variables are needed (i.e., Tseng, Costa, Jung, and Choudhury [37] preferably with larger datasets. One such approach that can potentially drive this field toward a bio-esignature of substance use behaviors is mobile sensing [38]. Furthermore, more diverse and accurate repeated measures of executive cognitive functioning may further our predictive capabilities. 

## 5. Conclusions

In this clinical sample, we found that continuous measures of mental health distress predict subsequent cognition, that continuous variability in cognition predict craving, and that continuous variability in craving predict relapse. Overall, within-subject variability distinguished relapse/no-relapse better than the group level summary statistics (except for baseline response speed). This informs us that mental health distress is a fairly immediate predictor of relapse and that the variability in cognition and craving are the best indicators of subsequent craving intensity and relapse, respectively. Although limited by sample size, our findings increase the current knowledge about the temporal relationships in substance use craving, relapse, mental health, and neurocognitive factors in clinical populations and will advance our future ability to identify not just who, but rather when patients are at risk for substance use and enable just-in-time and more personalized interventions. 

Our findings may inform clinical practice by helping clinicians focus on performing brief repeated measures of symptoms typically experienced during substance use treatment and using the levels and variability of these measures to understand the current level of risk of symptom deterioration to inform treatment intensity and mode. It is noteworthy that substance use relapse is often surprising for the patients and health care providers, and our findings are hopefully a step toward ameliorating this situation.

## Figures and Tables

**Figure 1 brainsci-12-00957-f001:**
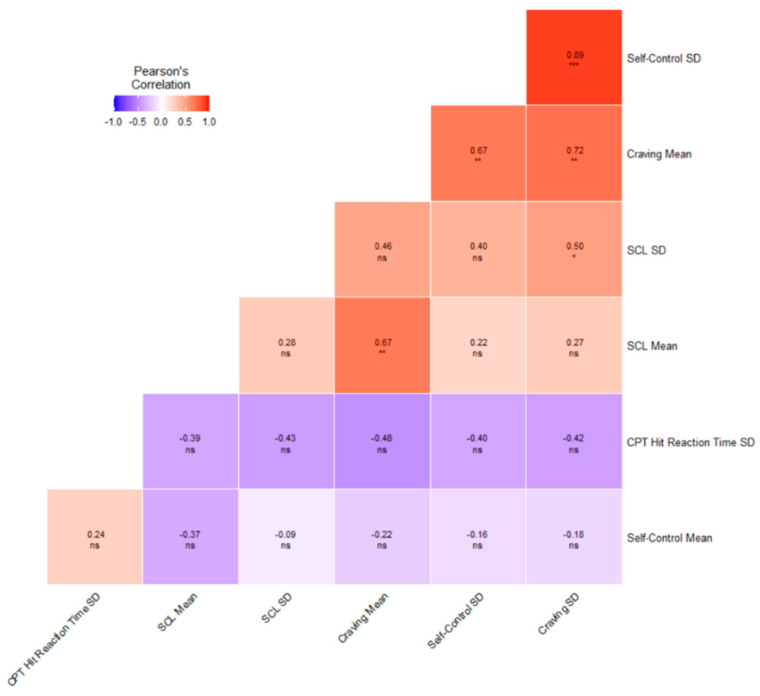
The Pearson product–moment correlation coefficients between the repeated measures summary scores and baseline variability in reaction time (CPT.HRT.SD) among the predictor variables. The mean and standard deviations for the repeated measures variables were calculated across occasions for each participant. The CPT HRT SD was measured once at the baseline. SCL—Hopkins Symptom Check-list 5 item version, CPT HRT SD—Conners Continuous performance test 3rd version, standard deviation of hit reaction time. Note: NS = No significance; Symbols *, **, and *** denote the significance levels at *p* < 0.05, *p* < 0.01, and *p* < 0.001, respectively, within-subject analyses.

**Figure 2 brainsci-12-00957-f002:**
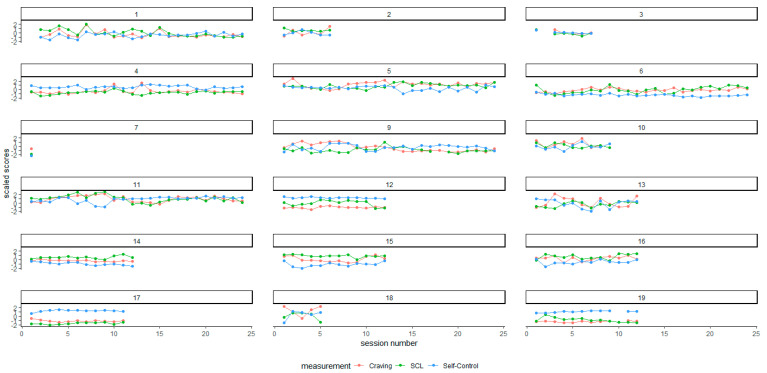
The repeated measures for participants 1–19, standardized scores for comparison (standardized using the mean and standard deviations across occasions) for craving, Hopkins Symptom Check List 5-item version (SCL) and self-control. Participant 8 was removed due to missing data, participant 7 only reported data on one occasion post baseline.

**Table 1 brainsci-12-00957-t001:** The group level descriptive statistics of the baseline and repeated measures for the full sample and between group comparisons for the relapsed and non-relapsed participants. WAIS: Wechsler Adult Intelligence scale 4th ed., D-KEFS: Delis-Kaplan Executive Function System, CPT: Conners Continuous Performance test, 3rd ed. Wilcoxon–Mann–Whitney test for between group differences for relapse. Effect sizes (r) are reported as absolute values. Symbol * denotes significance level at *p* < 0.05. WAIS- and CPT scores are T-scores. D-KEFS scores are scaled, except for the raw score for verbal fluency: category switching.

	Full Sample		Relapse						
			*YES*		*NO*				
	*Median*	*IQR*	*Median*	*IQR*	*Median*	*IQR*	*r*	*p*-Value	
**WAIS Baseline**									
Working Memory Index	85	9.5	82	10	86.5	14	0.235	0.327	
Verbal Comprehension Index	91	20.5	81	14	94	17	0.349	0.139	
Perceptual reasoning: Block Design	8	3.5	8	2	9	4.25	0.206	0.392	
Processing Speed: Coding	8	2	8	2	8.5	2.25	0.010	1.000	
Visual Perception: Picture completion	10.5	3	9	4	11	3	0.143	0.606	
**D-KEFS Baseline**									
Cognitive Flexibility: Trail Making test (TMT) Scaled Score	8	4	9	2.5	5.5	5.5	0.333	0.185	
Color-Word Interference Test (Stroop) Scaled Score	7	3.5	8	2.5	5.5	3	0.264	0.267	
Cognitive flexibility: Verbal Fluency—Category switching	9	3.5	8	2.5	9	3.5	0.236	0.324	
**CPT Baseline**									
Inattentiveness: Commissions	58.5	13.75	60	10.5	57	15.5	0.214	0.389	
Inattentiveness: Omissions	47	3.5	45	3.5	48	2.5	0.349	0.152	
Impulsivity: Perseverations	48	11.75	48	5.5	48	18	0.155	0.541	
Vigilance HRT-ISI	49	8	48	4	49	10.5	0.107	0.683	
Hit reaction Time/Response Speed: HRT	40.5	6	36	5.5	42	3	0.515	0.032	*
Response Speed Consistency: HRT SD	47.5	10.25	45	3	53	11.5	0.567	0.018	*

**Table 2 brainsci-12-00957-t002:** The group level statistics for repeated measures. SCL: Hopkins Symptom Check List—5 item version. The mean and SD are across sessions for each participant. Median and IQR of Mean and SD are across participants. The Wilcoxon–Mann–Whitney test for between group differences for relapse. Effect sizes (r) are reported as absolute values.

	Full Sample		Relapse					
			*YES*		*NO*			
	*Median*	*IQR*	*Median*	*IQR*	*Median*	*IQR*	*r*	*p*-Value
**Craving**								
Mean across occasions	32.98	15.61	25.79	28.35	25.79	28.35	0.416	0.085
SD across occasions	15.29	14.68	12.93	11.51	12.93	11.51	0.402	0.107
**Self-Control**								
Mean across occasions	60.94	29.59	64.50	24.50	64.50	24.50	0.245	0.319
SD across occasions	13.45	8.17	10.57	4.18	10.57	4.18	0.308	0.223
**SCL**								
Mean across occasions	10.57	3.61	9.33	1.45	9.33	1.45	0.374	0.124
SD across occasions	1.83	0.88	1.63	1.21	1.63	1.21	0.308	0.223

**Table 3 brainsci-12-00957-t003:** The mixed effects models (LMM and GLMM) with observations nested within subjects. SCL: Hopkins Symptom Check-list 5 item version, CPT HRT SD: Conners Continuous performance test 3rd version, standard deviation of hit reaction time. Symbol * denotes the significance level at *p* < 0.05. Beta-coefficients, SE: Standard error, R2: R-squared, AIC: Akaike information criterion, Adj.ICC: Adjusted Intraclass coefficient.

	Beta	SE	R2	AIC	Adj ICC	*p*-Value
**Model 1: Self-control *t + 1***			0.74	1251.8	0.71	
Intercept	67.45	29.67				0.036
SCL	−2.10	0.61				7.94 × 10^−4^ *
SCL 3-session rolling SD	−1.33	1.42				0.351
CPT HRT SD (Baseline)	0.38	0.55				0.493
**Model 2: Craving *t + 1***			0.56	1321.6	0.44	
Intercept	79.83	27.60				0.086
Self-control	0.03	0.11				0.787
Self-control 3-session rolling SD	0.55	0.24				0.023 *
SCL	0.59	0.83				0.483
SCL 3-session rolling SD	−2.42	2.00				0.229
CPT HRT SDSD (Baseline)	−1.08	0.45				0.033 *
**Model 3: Relapse *t + 1***			0.74	48.7	0.18	
Intercept	−6.69	3.45				0.053
Self-control	−0.06	0.04				0.135
Self-control 3-session rolling SD	−0.18	0.12				0.143
SCL	0.43	0.29				0.132
SCL 3-session rolling SD	−0.43	0.59				0.470
Craving	0.01	0.03				0.805
Craving 3-session rolling SD	0.18	0.10				0.061
**Model 4: Relapse *t + 1***			0.36	48.4	0.18	
Intercept	−5.10	1.27				5.79 × 10^−5^ *
Craving 3-session rolling SD	0.10	0.04				0.020 *

## Data Availability

The data presented in this study are not publicly available due to confidentiality.

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
