# Peer review of "Predicting Relapse in Substance Use: Prospective Modeling Based on Intensive Longitudinal Data on Mental Health, Cognition, and Craving"

_brainsci, 2022, doi:10.3390/brainsci12070957_

Round 1
Reviewer 1 Report
This study address an important research problem on relapse in substance use within a limited sample population of patients in Norwegian. It is very interesting on how authors used predictive modes based on longitudinal data for mental health, cognition and craving in addressing the variables in question. Although this is a very interesting study, I have number of concerns:
Abstract and Introduction
Correct the referencing style.
Abstract must be structured in one paragraph.
Keywords are not listed.
There are grammatical errors in the manuscript.
The first and second paragraphs in the introduction do not interlink. Also, the problem under investigation can be identified from the introduction but the whole introduction section must be rearranged in order to convey the message to the readers.
The second statement under aim could be moved to methodology.
Methodology
The methodology applied is well detailed however, the following must be addressed:
Table 1 should be numbered and labelled on top.
Since the “diagnoses were made by their treatment responsible psychologist or physician and not reconsidered for this study” what is the relevance of table 1 in this study?
Results
Summary statistics of baseline and repeated measures are available in Error! Reference source not found.. What did you wanted to indicate here?
All the tables in this section are numbered and labelled below. Please see my previous comment- name and number all the tables above them, and should be properly cited in-text.
Discussion
Line 306-308 has been mentioned under introduction. Consider removing it, it does not add any new information.
“This adds to the current knowledge from crossectional studies of substance use populations, where perceived self-control has been found to be associated with mental distress (Abel et al., 2018)”. Are there any other studies that support this current knowledge? You stated one and mentioned “cross-sectional studies”. The same comment applies also here: “This has previously primarily been found in longitudinal studies with fewer measuring points than our study (e.g., Felton et al. (2019)”.
Line 333- could you briefly mention those previous findings?
The methodological limitations are mentioned but the strengths are not stated. In terms of methodology and contribution to the body of knowledge, what is new in this study? This must be a concluding paragraph of your discussion.
Conclusion
“Future research” must be the last parts of your discussion, without giving it a heading.
Author Response
Thank you for your thorough read and constructive suggestions.
Changes are marked with Track changes in the document.
Abstract
1. Abstract must be structured in one paragraph.
Response:
This was rightfully pointed out by both reviewers and has now been changed. I hope our rewritings meet your requirements
Introduction
2. The first and second paragraphs in the introduction do not interlink.
Response
Thank you for pointing this out, we have now performed a significant reorganization of these sections, now in lines 41-66. We believe this has improved the overall quality and coherency of these sections.
3. Also, the problem under investigation can be identified from the introduction but the whole introduction section must be rearranged in order to convey the message to the readers.
Response:
We have now done a considerable rewrite and reorganization of the first part. Reviewer 2 wanted it shorter and more pointed, and we have tried to accommodate both concerns. We have taken out some of the more general definitions and understanding of craving and tried to be more explicit concerning the temporal relationships between the variables under study. These changes are in lines 43-104.
4. The second statement under aim could be moved to methodology.
Response
Done. Is now found under ‘Statistical analyses’.
Materials and Methods
5. Table 1 should be numbered and labelled on top.
Response
Table 1 has now been moved to Appendix 1 with correct placement of caption.
6. Since the “diagnoses were made by their treatment responsible psychologist or physician and not reconsidered for this study” what is the relevance of table 1 in this study?
Response
Agreed, it was primarily included to illustrate the severity of substance use in the population, and for comparability with other studies. We did however remove it altogether and referred to this in appendix 1.
Results
7. Summary statistics of baseline and repeated measures are available in Error! Reference source not found.. What did you wanted to indicate here?
Response
This is an automatically generated Word-error message due to hyperlinked Table. This has now been corrected for Table 1 and 2.
8. All the tables in this section are numbered and labelled below. Please see my previous comment- name and number all the tables above them and should be properly cited in-text.
Response
We have now made these corrections.
Introduction
9. Line 306-308 has been mentioned under introduction. Consider removing it, it does not add any new information.
Response
We removed the reiteration of the aim.
10. “This adds to the current knowledge from crossectional studies of substance use populations, where perceived self-control has been found to be associated with mental distress (Abel et al., 2018)”. Are there any other studies that support this current knowledge? You stated one and mentioned “cross-sectional studies”.
Response
We have now added the study by Garke (2021) in line 384 even though it had a slightly broader scope. We could also have repeated some older references mentioned in the Abel et al. study, but we have chosen not to do this due to methodological difficulties in some of these studies. Hope this is according to your expectations.
11. The same comment applies also here: “This has previously primarily been found in longitudinal studies with fewer measuring points than our study (e.g., Felton et al. (2019)”.
Response
There is a scarcity in longitudinal research addressing this, especially in more intensive repeated measure designs such as ours. At this point we hope that the Felton study as well as the later reference to McKee, are sufficient.
12. Line 333- could you briefly mention those previous findings?
Response
This has now been done in lines 392-394
13. The methodological limitations are mentioned but the strengths are not stated. In terms of methodology and contribution to the body of knowledge, what is new in this study? This must be a concluding paragraph of your discussion.
Response
We hope this has now been better addressed in lines 403-407.
14. “Future research” must be the last parts of your discussion, without giving it a heading.
Response
This has now been rectified.
Reviewer 2 Report
Abstract
1. Instead of 3 separate sections, the abstract should be one paragraph long.
2. It is advised to provide some statistical results with numbers (such as r correlation and p value) in the abstract because the statistical analysis was utilized as the primary method of analysis all across the study.
Keywords
3. At least 3 specific keywords for the paper must be provided.
Introduction
4. The introduction contains excessive detail. Please think about keeping it to one page and being concise.
Participants
5. Instead of writing "The results from the working memory training will be reported elsewhere." at line 104, please replace them with brief descriptions.
6. To better understand the characteristics of the sample, please provide more background information about the participants.
Measures
7. The setting of the measurements is unclear. Because executive function and attention outcomes might be affected by an improper environment, please provide more details.
Results
8. In Tables 2 and 3, you need to indicate what the symbol * refers to.
9. Please report the effect sizes for all the statistically significant results and discuss them.
10. Please remove all duplicate results from Figure 1 and include the p values where statistically significant results were found.
11. What is the difference between T+1 in Table 3 and t+1 in line 286?
12. The definitions of all the abbreviations used in each table and figure should be included in the table and figure notes.
13. The format for reporting the decimal places is inconsistent. It is typically preferred to report other results with two or three points and the p value with three points.
Discussion
14. Due to the small sample size, you need to properly mention this issue as a limitation with a suggestion for further study.
15. There are some links between the limitations and the further research parts. It is better to combine those two parts into a single paragraph and place them at the end of the Discussion section.
Conclusions
16. You need to summarize the study by including the main findings in this section.
References
17. DOIs for nos. 10 and 20 need to be rechecked, and in line 501, "World Health, O" should be corrected.
Others
18. Lines 165, 186, 224, 242, 250, and 281, "Error! Reference source not found" should be fixed.
19. Please review the entire text between lines 381 and 415 to ensure that any extraneous information has been eliminated.
Author Response
Thank you for your thorough read and constructive feedback, changes in the manuscript are identified using track changes.
1. Instead of 3 separate sections, the abstract should be one paragraph long.
Response
This was rightfully pointed out by both reviewers and has now been changed.
2. It is advised to provide some statistical results with numbers (such as r correlation and p value) in the abstract because the statistical analysis was utilized as the primary method of analysis across the study.
Response
This is a very reasonable comment. However, because the effects described in the abstract are complex (outcome predicted by temporal variability, correlations between within-subject variability and outcomes), it is challenging to provide detailed statistics in the short format allowed in the abstract. For example, ‘Analyses revealed strong correlations between within-subject variability (sd) in mental health, self-control, and craving’ refers to 3 different correlations (MHxCraving, MHxSelf-Control, CravingxSelf-Control). The results from the modeling it is even more complex to report in the abstract, not the least due to the prospective nature of the analyses. We have therefore chosen not to do this. If this is not satisfactory, we will remove some parts of the abstract to accommodate this.
4. At least 3 specific keywords for the paper must be provided.
Response
Thank you for discovering this. We have now added Keywords
Introduction
5. The introduction contains excessive detail. Please think about keeping it to one page and being concise.
Response
We have now done a considerable rewriting of Introduction. Reviewer 2 wanted it shorter and more pointed, and we have tried to accommodate both concerns. We have removed some of the more general definitions and statements surrounding craving and tried to be more explicit concerning the temporal relationships between the variables under study. Changes throughout the introduction are marked with track changes.
Materials and Methods
6. Instead of writing "The results from the working memory training will be reported elsewhere." at line 104, please replace them with brief descriptions.
Response
We have not yet analyzed these findings, but studies of the effects of working memory training have at best been modest. We have removed this sentence but have mentioned possible effects of working memory training in the Discussion (lines 423-434).
7. To better understand the characteristics of the sample, please provide more background information about the participants.
Response
We added some descriptions in lines 107-109 and moved the table describing the diagnoses to Appendix 1. Lines 240-242 describe the age and sex characteristics of participants. The new Table 1 (previously table 2) gives detailed descriptive statistics about the baseline and repeated measurements, for the full sample and for relapse/non-relapse groups.
8. The setting of the measurements is unclear. Because executive function and attention outcomes might be affected by an improper environment, please provide more details.
Response
This is an important point for replicability and clinical context. For the readers, this information has now been added under the section ‘measures’ in lines 123-127.
Results
9. In Tables 2 and 3, you need to indicate what the symbol * refers to.
Response
These are now Tables 1 and 2, and this has been carried out.
10. Please report the effect sizes for all the statistically significant results and discuss them.
Response
We have now added information about the effects of the significant predictor variables on the output variables in lines 329-349 by reporting predicted changes in continuous outcomes (Craving, Self-Control and SCL/Mental Health) for selected changes in the predictor values considered meaningful. We also calculated effects using odds ratio for the binary outcome of relapse. There is some complexity in reporting effect sizes in prospective predictions in mixed effects models, but we hope this has now been done to your satisfaction. The effects have been further discussed in the discussion section lines 362-371.
In addition, we have also added Appendix 2-4 to illustrate the variability in the significant input variables.
11. Please remove all duplicate results from Figure 1 and include the p values where statistically significant results were found.
Response
This has now been done.
12. What is the difference between T+1 in Table 3 and t+1 in line 286?
Response
There is no difference, we have now kept the notation t+1 and removed all mention of T+1. Thank you for pointing this out.
13. The definitions of all the abbreviations used in each table and figure should be included in the table and figure notes.
Response
This has now been done throughout.
14. The format for reporting the decimal places is inconsistent. It is typically preferred to report other results with two or three points and the p value with three points.
Response
We agree that it is inconsistent, we would like the editor to give us guidance on how many significant digits should be reported and how, as we find little guidance on this in the author instructions. We will comply as soon as we receive this input.
Discussion
15. Due to the small sample size, you need to properly mention this issue as a limitation with a suggestion for further study.
Response
This should be covered in line 405 and 431..
16. There are some links between the limitations and the further research parts. It is better to combine those two parts into a single paragraph and place them at the end of the Discussion section
Response
This has now been done by merging these two parts.
17. Lines 165, 186, 224, 242, 250, and 281, "Error! Reference source not found" should be fixed.
Response
This has now been fixed, and the correct table numbers have now been inserted.
18. Please review the entire text between lines 381 and 415 to ensure that any extraneous information has been eliminated.
Response
All extraneous and less relevant information has been removed.
References
DOIs for nos. 10 and 20 need to be rechecked, and in line 501, "World Health, O" should be corrected.
All of these have now been checked and corrected.
Round 2
Reviewer 2 Report
Abstract
1. The predictive analysis and results should be the key findings in this study, according to the title. However, there is only a small part indicating that important information, while the last part of the abstract takes four lines to describe the benefits. Please be considerate in rewriting the abstract by reducing the benefit part and using that space to emphasize more of the analysis and result parts.
Introduction
2. It doesn't seem necessary to include the word “Aims” in the subheading. Please remove it.
3. There are some symbols (e.g., hyphen and underscore) that need to be fixed in this section, such as “-the” in line 47 and “are- _often” in line 53. There are also some writing errors. Please reread this Section and make the necessary corrections.
Participants
4. Since this study is involved with clinical trials, please be considerate in using the term “participants” instead of “subjects” throughout the manuscript.
Reference: Chalmers I. People are “participants” in research. Further suggestions for other terms to describe “participants” are needed. BMJ. 1999 Apr 24;318(7191):1141. doi: 10.1136/bmj.318.7191.1141a. PMID: 10213744; PMCID: PMC1115535.
5. Since the sample size in this study is so small, it cannot represent the whole population. Please provide more background information about the participants (e.g., state, city, region, or nationality).
Statistical analysis
“the Wilcoxon-rank test was used to compare means”
6. The Wilcoxon signed-rank test is a nonparametric test to compare two matched or paired datasets. However, the study compared between the relapsing and non-relapsing groups, which are two independent groups (in Table 1). In this case, the Mann-Whitney U test should then be used.
7. By considering using a nonparametric test, the descriptive statistic should be presented as the median and interquartile range.
8. Effect sizes for nonparametric data, such as the Mann-Whitney and the Wilcoxon tests, should be estimated and reported with r, r2, or η2.
Reference: Fritz CO, Morris PE, Richler JJ. Effect size estimates: current use, calculations, and interpretation. J Exp Psychol Gen. 2012 Feb;141(1):2-18. doi: 10.1037/a0024338. Epub 2011 Aug 8. Erratum in: J Exp Psychol Gen. 2012 Feb;141(1):30. PMID: 21823805.
Results
9. Line 242 should be “7 females”.
10. At this point, please consider reporting all results with two decimal places (e.g., r = 0.89) and all p values with three decimal places (e.g., p = 0.030) in all tables, figures, and their corresponding text in the same format to improve the consistency of the manuscript.
11. For Tables 1 and 2, all abbreviations and symbols should be indicated under the table, as per the example below.
“Note: WAIS = The Wechsler Adult Intelligence Scale. Symbols * and ** denote significance levels at p < 0.05 and p < 0.01, respectively.”
12. For Table 1, the table header should be modified. Here is an example:
|
Baseline |
Mean (SD) |
p value |
||
|
All participants (N=19) |
Relapsing group (n=7) |
Non-relapsing group (n=12) |
||
|
Working memory index |
87.95 (11.30) |
83.86 (5.30) |
90.33 (13.30) |
0.030* |
13. In the repeated results in Table 1, the numbers in the parenthesis reported which values? Since there is an SD row for each repeated test.
14. Figure 1 should be mentioned in the fourth paragraph of the Results Section.
15. In the fourth paragraph of the Results Section, the results should specify if a correlation is positive or negative.
16. What are the background colors in Figure 1 referring to and what for?
17. The study stated “Ns p>=0.05, *p<0.05,**p<0.01, and ***p<0.001.” in line 280.
My suggestion is “Note: NS = No significance. Symbols *, ** and *** denote significance levels at p < 0.05, p < 0.01, and p <0.001 respectively.”
18. “SDSD” (lines 277 and 278) should be spelled “SD”.
19. All 18 images can be rearranged as 6 x 3 or 3 x 6 to remove the empty area at the right-side corner of Figure 2.
20. For Figure 2, rather than indicating “measurement_value” at the vertical axis, please specify an actual unit (e.g., scaled score).
21. In Figure 2, the numbers 25 and 0 are so close together on the horizontal axis. This might be confused with 250.
22. Please indicate what numbers 1 to 19 in Figure 2 refer to, such as “Participant 1” and so on.
23. According to Figure 2, the comparison result reported between lines 288 and 291 is quite short and does not seem to contribute much to the study. Please consider reporting more details of the comparative outcomes from this figure.
24. In the p value column in Table 2, there are two exponential results. Is it possible to replace them with actual p value numbers? If not, please substitute another symbol, such as the multiplication sign, for the * after the numbers 7.94 and 5.79.
Discussion
“Because we were able to identify significant effects with a relatively small sample, it is reasonable to believe that these effects are indeed high.” between lines 412 and 414.
25. This point should be interpreted with caution since the study does not have proof. Please consider revising this sentence.
Conclusions
26. This section only presents the advantages of the study, not any major interpretations or conclusions. Please highlight the significant results of the study and briefly interpret them in the first part of this section. Also, it is advisable to summarize the study without citing any references.
Overall
27. Please correct “Error! Reference source not found” in lines 200 and 270.
28. There are too many forms of Self-Control e.g., SelfC in line168, SelfC_ in line 272, SC in line 286, and Self C in Table2. Please only use one form to indicate one term, or else you can use a full word to avoid being misunderstood. To avoid this problem, please double-check all other abbreviations in the manuscript.
